# Modeling of DNA binding to the condensin hinge domain using molecular dynamics simulations guided by atomic force microscopy

Hiroki Koide[1], Noriyuki Kodera[2], Shveta Bisht[3], Shoji Takada[1], Tsuyoshi Terakawa[1,4]*

1 Department of Biophysics, Graduate School of Science, Kyoto University, Kyoto, Japan, 2 Nano Life Science Institute (WPI-NanoLSI), Kanazawa University, Kanazawa, Japan, 3 Cell Biology and Biophysics Unit, Structural and Computational Unit, European Molecular Biology Laboratory (EMBL), Heidelberg, Germany, 4 PREST, Japan Science and Technology Agency (JST), Kawaguchi, Japan

* terakawa@biophys.kyoto-u.ac.jp

**Data Availability Statement:** All data that support the findings of this study are available within the manuscript and the supplementary information.

## Abstract

The condensin protein complex compacts chromatin during mitosis using its DNA-loop extrusion activity. Previous studies proposed scrunching and loop-capture models as molecular mechanisms for the loop extrusion process, both of which assume the binding of double-strand (ds) DNA to the hinge domain formed at the interface of the condensin sub-units Smc2 and Smc4. However, how the hinge domain contacts dsDNA has remained unknown. Here, we conducted atomic force microscopy imaging of the budding yeast condensin holo-complex and used this data as basis for coarse-grained molecular dynamics simulations to model the hinge structure in a transient open conformation. We then simulated the dsDNA binding to open and closed hinge conformations, predicting that dsDNA binds to the outside surface when closed and to the outside and inside surfaces when open. Our simulations also suggested that the hinge can close around dsDNA bound to the inside surface. Based on these simulation results, we speculate that the conformational change of the hinge domain might be essential for the dsDNA binding regulation and play roles in condensin-mediated DNA-loop extrusion.

## Author summary

Condensin is a protein which plays important roles in proper chromosome condensation and segregation. Recent studies have suggested that the chromosome condensation is driven by a DNA loop extrusion activity of condensin and that binding of the condensin hinge domain to DNA underlies this activity. However, the structural model of the hinge/ DNA complex has not been available, which has limited our understanding of how condensin extrudes DNA loops. In this study, we performed high speed atomic force microscopy (AFM) imaging of budding yeast condensin complexes, conducted molecular dynamics simulations with constraints derived from the AFM image, and modeled the

**Funding:** This work was supported by PRESTO grant of Japan Science and Technology Agency (JPMJPR19K3; to T.T.; https://www.jst.go.jp/EN/), Grant-in-Aid for Scientific Research of Japan Society for the Promotion of Science (B; 19H03194; to T.T.; https://www.jsps.go.jp/english/index.html), Grant-in-Aid for Scientific Research on Innovative Areas of Japan Society for the Promotion of Science (Molecular engine; 19H05392; to T.T.; https://www.jsps.go.jp/english/index.html), Grant-in-Aid for Scientific Research on Innovative Areas of Japan Society for the Promotion of Science (Chromatin potential; 19H05260; to T.T.; https://www.jsps.go.jp/english/index.html), and the CREST grant of Japan Science and Technology Agency (JST) (JPMJCR1762; to S. T.; https://www.jst.go.jp/EN/). The funders had no role in study design, data collection and analysis, decision to publish, or preparation of the manuscript.

**Competing interests:** The authors have declared that no competing interests exist.

structures of the hinge domain with the open conformation and the hinge/DNA complex. The simulation results suggest that opening and closing of the hinge domain regulate its binding to DNA. This regulation might be relevant to the molecular mechanisms of DNA-loop extrusion.

## Introduction

Proper chromosome condensation before mitosis is essential to ensure that eukaryotic genomes are correctly distributed into the two daughter cells during cell division [1–3]. The condensin complex, a member of the Structural Maintenance of Chromosomes (SMC) protein family, is a key player for chromosome condensation. Indeed, purified condensin and topo-isomerase II are sufficient to induce the formation of cylindrical chromatids *in vitro* [4,5]. Recent single-molecule experiments revealed that the budding yeast condensin is a molecular motor [6] that extrudes a DNA loop [7,8]. However, the detailed molecular mechanism of DNA-loop extrusion by condensin and other members of the SMC protein family remains unclear [8–10] and has been an ambitious target of computational modeling studies [11–13].

The budding yeast condensin is composed of five subunits called Smc2, Smc4, Brn1, Ycg1, and Ycs4 [14]. Each of Smc2 and Smc4 forms an anti-parallel coiled-coil [15] that brings together N-terminal and C-terminal regions to create a globular ATPase 'head' domain at one end [16]. The other ends of Smc2 and Smc4 interact with each other to form the doughnut-shaped 'hinge' domain [17,18]. Brn1 interacts with the head domains of Smc2 and Smc4, closing the condensin ring [16,19]. Ycg1 and Ycs4 are largely composed of HEAT repeats and bind to the central region of Brn1. Ycg1 and potentially also Ycs4 bind double-strand (ds) DNA, and the crystal structure of the Ycg1/Brn1/dsDNA complex is available [16,20]. In addition, biochemical experiments and atomic force microscopy (AFM) imaging suggest that the hinge domain binds strongly to single-strand (ss) DNA [17,21–24] and weakly to dsDNA [18,25], respectively.

The two coiled-coils, one from Smc2 and the other from Smc4, emanate from the hinge domain. Previous studies revealed two conformations of the hinge domain; i) a 'closed' conformation in which the two coiled-coils are closely juxtaposed and ii) an 'open' conformation in which the coiled-coils are further apart [11,18,25–29]. Although these conformations have been structurally characterized, there is no clear understanding about the underlying mechanism of this transition. For budding yeast condensin, the structures of the closed hinge in the closed conformation were reported [18,25], while the open hinge conformation has been observed only in AFM imaging. The binding of dsDNA to the hinge domain have been proposed to regulate the inter-head engagement at the other ends of the coiled-coils, promoting ATP hydrolysis [21,30]. Previous studies have proposed 'scrunching' [25] and 'loop capture' models [31] for the molecular mechanism of the DNA loop extrusion process. The two models are not mutually exclusive, and both assume dsDNA binding to the hinge domain. However, no structural information about the interaction of the hinge with dsDNA is available.

In this study, we conducted AFM imaging of the budding yeast condensin holo-complex, and used the data to guide coarse-grained (CG) molecular dynamics (MD) simulations [32,33] to predict the dsDNA-binding surface of the condensin hinge domain in both, open and closed conformations. First, we demonstrated that our simulations correctly predict the Ycg1/dsDNA complex structure [20], validating our CG model. Next, we conducted high-speed (HS)-AFM imaging of the whole condensin complex to obtain open hinge images [25,34]. Using this data to constrain CGMD simulations [35], we obtained a structural model for the open hinge

conformation. Finally, we predicted dsDNA binding surfaces on the hinge using CGMD simulations. These simulations identified a transient dsDNA binding surface on the inside and outside of the hinge domain, consistent with the previous biochemical [24] and HS-AFM [25] studies. It was also suggested that the hinge opening and closing regulate the dsDNA binding to the hinge domain. From these results, we speculate that structural changes in the hinge domain caused by ATP hydrolysis and the accompanying dsDNA binding might contribute to the chemo-mechanical coupling in the condensin DNA-loop extrusion.

## Materials and methods

### Prediction of the Ycg1/dsDNA and Ycs4/dsDNA complex structures

Previous studies have shown that electrostatic interactions dominate protein/dsDNA interactions [36]. It was also demonstrated that CGMD simulations that consider only electrostatic interactions and excluded volume effect as interactions between proteins and dsDNA could predict protein/dsDNA complex structures with a certain degree of accuracy [37–39]. In this study, to validate the model, the Ycg1/dsDNA complex structure, whose crystal structure has already been solved [20], was assessed by CGMD simulations. In the crystal structure of the Ycg1/Brn1/dsDNA complex, the dsDNA fits into the Ycg1 binding cleft and is covered by Brn1. This topology makes conformational sampling of DNA binding to the Ycg1/Brn1 complex by CGMD simulations almost impossible. As Brn1, which binds to DNA with a few amino acid residues, would play a merely auxiliary role in DNA binding, we neglected Brn1 and conducted the blind prediction of the Ycg1/dsDNA complex structure. The 2.2Å root mean square displacement between DNA bound and unbound crystal structures suggests that the effect of protein flexibility to DNA binding is minor, if any. As a reference for binding strength, we also predicted the Ycs4/dsDNA complex structure.

Here, we provide a brief explanation of the AICG2+ protein model used in this study (Please refer to the original work [40] for detail). In this model, each amino acid is represented by one bead located on the $C_\alpha$ atom position. Consecutive amino acids are connected by elastic bonds. A sequence-based statistical potential is used to model bond angles and dihedral angles. An excluded volume interaction prevents two beads from overlapping each other. The native-structure-based contact potential (Gō potential) constrains the distance between amino acid pairs that contact each other in the native Ycg1 (PDB ID: 5OQN) and Ycs4 structures. To model the native Ycs4 structure, we performed homology modeling using *Chaetomium Thermophilum* Ycs4 structure (PDB ID: 6QJ4, 47.7% sequence identity) as a template (https://salilab.org/modeller/). Parameters in the AICG2+ model were decided so that the fluctuation of each amino acid in reference proteins reproduces that of all-atom simulations.

Next, we provide a brief explanation of the 3SPN.2C DNA model (Please refer to the original work [41] for detail). In this model, each nucleotide is represented by three beads located at the positions of base, sugar, and phosphate units. Neighboring sugar-phosphate and sugar-base are connected by virtual bonds. Bond angles and dihedral angles are restrained to their values in reference B-type dsDNA. An excluded volume interaction prevents any two beads from overlapping each other. Orientation-dependent attractive potentials are applied to base pairs, cross-stacking pairs, and intra-chain-stacking pairs. The parameters are decided so that the model reproduces several types of experimental data.

When a protein forms a complex with dsDNA, the electrostatic interaction plays an important role [36]. Instead of putting unit charges on the $C_\alpha$ atoms of charged residues, we placed partial charges on the $C_\alpha$ atoms of surface residues. The RESPAC algorithm [37] decided the partial charges of the surface residues to reproduce the electrostatic potential calculated for the all-atom structure model. The Debye-Hückel potential represents the interaction between

charge pairs. The potential is expressed as

$$V_{DH} = \sum_{i<j}^{N} \frac{q_i q_j}{4\pi\varepsilon_0\varepsilon_k r_{ij}} e^{-r_{ij}/\lambda_D} \tag{1}$$

where $q_i$ is charge, $\varepsilon_0$ is the electric constant, and $\varepsilon_k$ is the dielectric constant. The Debye length $\lambda_D$ is expressed as

$$\lambda_D = \sqrt{\frac{\varepsilon_0\varepsilon_k k_B T}{2N_A e^2 I}}, \tag{2}$$

where $k_B$ is the Boltzmann constant, $T$ is temperature, $N_A$ is the Avogadro's number, and $I$ is ionic strength. In addition to the electrostatic interaction, the excluded volume interaction is applied.

In the simulations, we put one protein molecule (Ycg1 or Ycs4) and five 18 base-pairs (bps) dsDNA molecules in the 200 Å × 160 Å × 200 Å box (Ycg1) and 210 Å × 170 Å × 210 Å box (Ycs4), respectively. The time evolution was modeled by Langevin dynamics with the friction coefficient and the time step set to 0.843 and 0.2, respectively. The monovalent ion concentration was varied as 100, 150, 200, 250, and 400 mM. We conducted the simulations using CafeMol version 3.2 [32] (https://www.cafemol.org).

## HS-AFM imaging of condensin holo-complexes

We conducted HS-AFM imaging to observe equilibrium dynamics of whole condensin complexes on bare mica surface in solution [42]. In HS-AFM imaging, the forces that a sample imposes on a probe (tapping-cantilever) are used to form an image of a three-dimensional shape of a protein surface.

We purified condensin holo-complexes as described previously [6]. Briefly, the five subunits of a condensin complex were co-overexpressed in *Saccharomyces cerevisiae* using galactose-inducible promoters. Cultures were grown at 30°C in URA/TRP dropout media containing 2% raffinose to $OD_{600}$ of 1.0. Expression was induced with 2% galactose for 8 hours. Then, cells were harvested by centrifugation, re-suspended in buffer A (50 mM Tris-HCl pH 7.5, 200 mM NaCl, 5% (v/v) glycerol, 5 mM β-mercaptoethanol, and 20 mM imidazole) containing 1× complete EDTA-free protease-inhibitor mix (Roche) and lysed in a Freezer-Mill (Spex). The lysate was cleared by two rounds of 20 min centrifugation at 45,000 × g at 4°C and loaded onto a 5-mL HisTrap column (GE Healthcare) pre-equilibrated with buffer A. The resin was washed with five column volumes of buffer A containing 500 mM NaCl; buffer A containing 1 mM ATP, 10 mM KCl and 1 mM MgCl₂; and then buffer A containing 40 mM imidazole to remove non-specifically bound proteins. Protein was eluted in buffer A containing 200 mM imidazole and transferred to buffer B (50 mM Tris-HCl pH7.5, 200 mM NaCl, 5% (v/v) glycerol, 1 mM DTT) using a desalting column. After the addition of EDTA to 1 mM, PMSF to 0.2 mM, and Tween20 to 0.01%, the protein was incubated overnight with 2 mL (bed volume) of pre-equilibrated Strep-Tactin high-capacity Superflow resin (IBA). The Strep-Tactin resin was packed into a column and washed with 15 resin volumes of buffer B by gravity flow. The protein was eluted with buffer B containing 5 mM desthiobiotin. The elute was concentrated by ultrafiltration and loaded onto a Superose 6 size exclusion chromatography column (GE Healthcare) pre-equilibrated in buffer B containing 1 mM MgCl₂. Peak fractions were pooled and concentrated to 4 μM by ultrafiltration.

We used a laboratory-built tapping-mode HS-AFM apparatus together with small cantilevers (BLAC10DS-A2, Olympus) designed for HS-AFM imaging (spring constant, ~0.1 N/m, resonant frequency in water, ~0.5 MHz; quality factor in water, ~1.5) [43]. We deposited 2 μL

of 1 nM condensin in buffer (40 mM Tris-HCl pH7.5, 125 mM KCl, 10 mM MgCl$_2$, 1 mM DTT, 4 mM ATP) onto the scanning mica stage of an HS-AFM apparatus. We conducted imaging at room temperature. The cantilever's free oscillation amplitude $A_0$ and set-point amplitude $A_s$ were set at 1–2 nm and ~0.9×$A_0$, respectively. The scan area was $150 \times 150$ nm$^2$, and the imaging rate was 150 ms/frame. To define a hinge angle, first, we found out the pixel with the highest height value in a hinge globular domain. We calculated the total height values of the parallelogram regions with a height of 10 pixels and a width of 5 pixels that extended radially from the highest point and selected two of them with local maximum values, respectively. The angle between the two lines passing through the center of these regions was defined to be hinge angle.

## AFM fitting simulations of the hinge domain

Similar to the earlier reports, in the current HS-AFM imaging, we consistently observed open hinge conformations [25]. To obtain the open hinge structure, we performed CG flexible-fitting MD simulations in which the restraint potential energy derived from the HS-AFM image [35] was imposed, stabilizing the open hinge structure.

The crystal structure of the hinge domain (PDB ID: 4RSI) was used as the initial structure. Using PyMol (https://pymol.org), we manually performed rigid body rotation and translation of Smc2 and Smc4 subunits, respectively, to roughly fit the AFM image.

Starting from this structure, we performed CG flexible-fitting MD simulations [35]. The method shares the same philosophy with the flexible-fitting to the cryo-electron microscopy data [44]. The settings were similar as for the Ycg1/dsDNA complex described above, except that the AFM potential was imposed. The AFM potential was defined by

$$V_{AFM}(\boldsymbol{R}) = \kappa N_B T(1 - c.s.(\boldsymbol{R})), \tag{3}$$

where $\boldsymbol{R}$ collectively represents protein coordinates, $k_B$ is the Boltzmann constant, $T$ is temperature (set to 300K), and $c.s.(\boldsymbol{R})$ is the similarity score (the cosine similarity, originally designated as a modified correlation coefficient) between the experimental AFM image and the pseudo-AFM image calculated from the simulations defined by

$$c.s.(\boldsymbol{R}) = \frac{\sum_p^{pixels} H_p^{(exp)} H_p^{(sim)}(\boldsymbol{R})}{\sqrt{\sum_p^{pixels} \left(H_p^{(exp)}\right)^2} \sqrt{\sum_p^{pixels} \left(H_p^{(sim)}(\boldsymbol{R})\right)^2}} \tag{4}$$

where $H_p^{(exp)}$ and $H_p^{(sim)}(\boldsymbol{R})$ are an experimental AFM image (a matrix listing all height values) and a pseudo-AFM image calculated from the simulated structure $\boldsymbol{R}$, respectively.

To decide the value of the parameter $\kappa$, we conducted 240 runs of $10^7$-steps simulations with $\kappa$ set to 10, 3000, and 10000, respectively. Then, to assess how the native structures were perturbed during the fitting, we calculated the standard nativeness score, Q-score, for intra-subunit and inter-subunit contacts. Q-score is defined as the ratio of the number of bead-bead contacts formed at a certain moment in simulations to the number of contacts formed in the native protein structure. The particle pairs with a distance less than native distance times 1.2 are considered to contact each other in CGMD simulations.

## Prediction of hinge/dsDNA complex structures

To predict dsDNA binding surfaces on the hinge domain, we placed the hinge fragment and five 18-bps dsDNA molecules in a simulation box (330 Å × 330 Å × 330 Å box for the open hinge and 255 Å × 255 Å × 255 Å box for the closed hinge) and performed CGMD

simulations. The last step of the AFM fitting simulation and the crystal structure (PDB ID: 4RSI) were used as the initial conformations for the open and closed hinge simulations, respectively. Initially, the DNA molecules were randomly placed in the box. The setup was the same as the CGMD simulations of the Ycg1 (or Ycs4)/dsDNA complex described above. In the open hinge simulations, the AFM potential with κ set to 3000 stabilized the open conformation.

## Results

### Blind prediction of the Ycg1/dsDNA complex

As a proof-of-concept that CGMD simulations can correctly predict DNA binding surfaces in the condensin complex, we first applied our approach to Ycg1. Since crystal structures of the Ycg1/Brn1/dsDNA complex have previously been reported (PDB ID: 5OQN), we can compare a blind prediction by simulations with the actual structure to assess the accuracy of simulations. For simplicity, we neglected Brn1 and conducted the blind prediction of the Ycg1/dsDNA complex structure.

We started the simulations with a single Ycg1 molecule at the center of the simulation box and five 18-bps dsDNA molecules randomly placed around Ycg1 (Fig 1A). In our CG model, the RESPAC algorithm [37] arranges the particle charges to reproduce the electrostatic potential calculated based on the all-atom charge arrangement. Therefore, we can predict DNA binding region more accurately than the conventional CG models [38,39]. After $1 \times 10^7$ steps of simulations, the dsDNA molecules spontaneously bound to the protein. To identify the dsDNA binding surface, we calculated the probability of protein-DNA contact for every amino-acid (Fig 1B). At the simulated salt concentration of 100 mM, residues with high DNA-contact probability values clustered in two regions on the protein surface referred to as sites I and II in Fig 1B. At higher salt concentration (400 mM; a condition of weaker electrostatic interactions), we only detected high contact probabilities for residues at the site I. Notably, the site I makes significant number of contacts with dsDNA in the available crystal structure (PDB ID: 5OQN).

To quantify the binding strength, we calculated the survival probability of dsDNA molecules bound to the sites I and II during consecutive simulation steps (Fig 1C). The dissociation rates of the sites I and II, estimated from the slopes of the fitting of the survival probabilities, were $1.2 \times 10^{-7}$ (steps$^{-1}$) and $1.0 \times 10^{-6}$ (steps$^{-1}$), respectively. Thus, in agreement with the experimental data, the CGMD simulations predicted the site I as the strongest dsDNA binding surface of Ycg1.

To get insight into how similar the structures in the simulations are to the crystal structure, we calculated Q-score for each simulation snapshot and plotted it as a function of time (S1 Fig). The Q-score represents how much fraction of the protein/DNA contacts in the crystal structure forms in each simulation snapshot. We considered that protein and DNA particles contacted when these were within 15Å. DNA particle identity was neglected to make this measure robust for DNA sliding. The plots reveal that, as the simulations proceed, Q-score increased as high as 0.9 and fluctuated between 0.5 and 0.9, i.e. 50–90% of the protein/DNA contacts in the crystal structure forms in the simulations.

Next, to obtain the representative structure, we performed conformational clustering to compare the predicted Ycg1/dsDNA complex structure with the crystal structure (PDB ID: 5OQN). First, we calculated the minimums of distances ($d_{min}^i$) between the $i$-th protein residue of the site I and every dsDNA particle. Second, we carried out the principal component analysis of structures using $d_{min}^i$ as coordinates. We then mapped each structure onto the first three principal component space. Finally, we counted the number of structures in a sphere with a radius of 10.0 centered on each structure in the unitless principal component space and

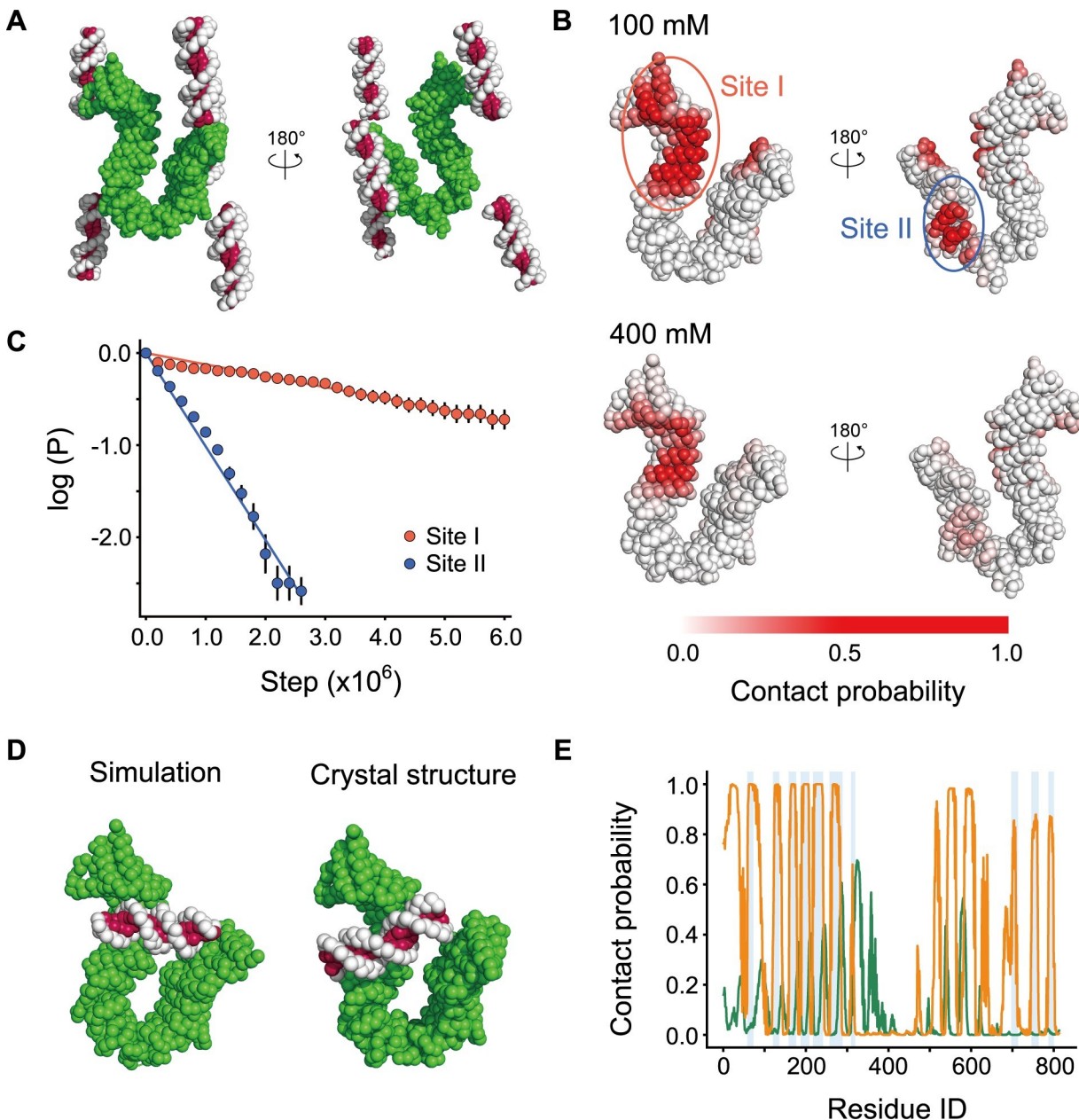

**Fig 1.** **(A)** The initial structure of CGMD simulations of the Ycg1/dsDNA complex. dsDNA is shown in white and magenta. Ycg1 is shown in green. **(B)** DNA contact probabilities mapped on the Ycg1 structure. Darker red particles contact with DNA with a higher probability. **(C)** A survival probability plot for the Ycg1/dsDNA binding. *P* represents the probability of DNA binding on Site I (red) or II (blue) after a certain duration. **(D)** Representative simulation (left) and the crystal (right; PDB ID: 5OQN) structures of the Ycg1/dsDNA complex. **(E)** Probabilities of each Ycg1 amino acid contacting with dsDNA. The orange and green lines represent the data from the simulations with the native and shuffled charge distribution. The vertical light blue lines represent the amino acid contacting dsDNA in the crystal structure.

defined the structure with the maximum value as the representative structure (Fig 1D left). The similarity between the representative structure and the crystal structure suggests that DNA binding surfaces can be successfully predicted using the CGMD simulations. Subtle differences between the two structures are most likely due to the simplicity of our CG model and the absence of Brn1 which, in addition to Ycg1, makes a few contacts with dsDNA in the crystal structure.

To quantitatively assess similarity between the representative structure and the crystal structure, we plotted the probability of each protein residue contacting to dsDNA (Fig 1E, orange line, 100 mM salt) and compared the probability with the DNA contact identified in the crystal structure (Fig 1E, blue bars). Most of the residues that contact dsDNA in the crystal structure were also predicted to contact dsDNA in the representative structure obtained by our simulation, except peaks between Ycg1 residues 450 to 650, which correspond to the site II. This correlation was strongly reduced when we randomly scrambled the charge arrangement (Fig 1E, green line), which suggests that the surface charge distribution dictates the dsDNA binding surface of Ycg1. This result is consistent with the fact that Ycg1 residues exclusively contact the dsDNA phosphate backbone [20].

## Prediction of the Ycs4/dsDNA complex

Next, we predicted the dsDNA binding surface on Ycs4, a potential dsDNA binder, to compare the binding strength of various proteins to dsDNA. As with the Ycg1 simulation, one Ycs4 was placed at the center of the simulation box and five 18-bps dsDNA molecules were randomly placed around it (S2A Fig). In $10^7$ steps of the simulations, dsDNA bound around the middle of the Ycs4 molecule with high probability (S2B Fig). Visual inspection revealed that dsDNA binds to two distinct surfaces on Ycs4 (the site I and II in S2C Fig). Future studies should address if dsDNA (weakly) binds to these surfaces.

## HS-AFM imaging of the condensin holo-complex

After having shown that the CGMD simulations can predict dsDNA binding surfaces in the condensin complex with a high degree of accuracy, we turned our attention to the Smc2/Smc4 hinge domain. To reveal the equilibrium hinge dynamics, we conducted HS-AFM imaging of the condensin holo-complex on bare mica surface in solution and observed 36 molecules. In representative AFM images (Fig 2A), the Smc2/Smc4 hinge domain, the coiled-coils and the ATPase head domains, as well as the two HEAT-repeat subunits Ycg1 and Ycs4 were observed, though, in most images, Smc2 and Smc4 head domains and HEAT-repeat subunits could not be clearly distinguished, similar to a previous HS-AFM study [25]. The HEAT-repeat subunits occasionally did not contact the head domains of Smc2 or Smc4 but stayed nearby (19/36) (S3 Fig), suggesting that they were merely bound to the complex by the disordered Brn1 subunit, which was not visible in AFM images. The transient detachment of one of the HEAT-repeat subunits is consistent with the recently proposed flip-flop model [28].

By HS-AFM imaging, we observed transitions between open and closed hinge conformations (Fig 2). Condensin with a closed hinge had a rod-like appearance, consistent with recent crystal and cryo-electron microscopy structures [26,28]. In the open conformation, where the coiled-coils are apart, we frequently observed a kink in the Smc2/Smc4 coiled-coils (25/36), which presumably corresponds to the 'elbow' region (the purple arrow in Fig 2A), though we did not observe the structure in which the hinge domain folds back and contacts the coiled-coils, as observed in the recent cryo-EM study [28]. To quantify the hinge dynamics, we plotted the time trajectory of hinge angle (Fig 2B). The plot shows that, in most of the time, the fluctuations in hinge angle remained below ~40° and only occasionally exceeded to 50–80° before spontaneously returning to values below ~40°. We observed the variation of angle in a single time trajectory, excluding the possibility that different angles arise due to orientations of the samples. Fig 2C shows the frequency distribution of hinge angle calculated from 15954 images of 36 molecules. The fitting of this distribution required at least two Gaussian distributions, which indicated that there are at least two conformational states; a stable closed conformation and a transient open conformation. These results reconcile the two conformations

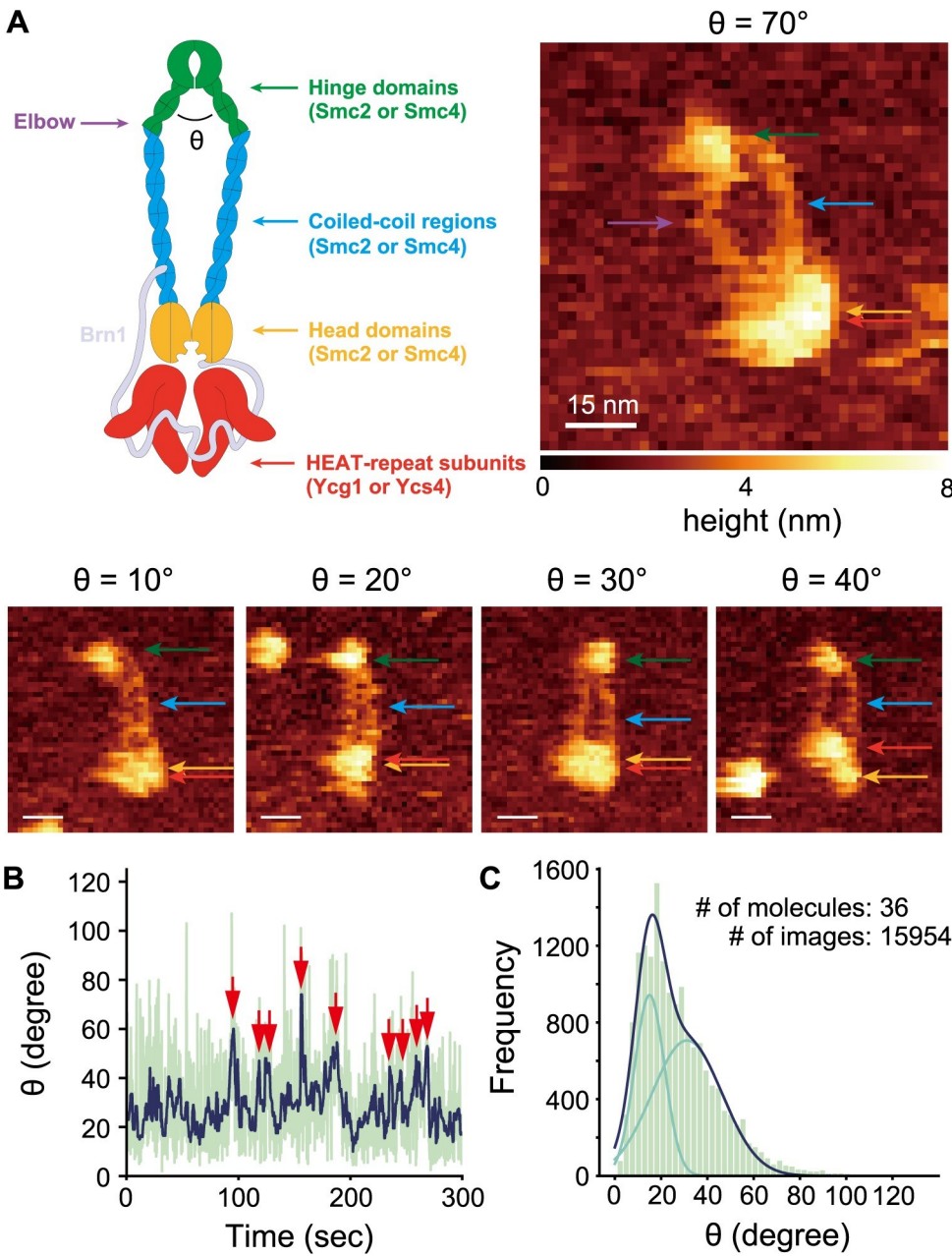

**Fig 2. (A)** The cartoon and the representative AFM images of the condensin holo-complex with various hinge angles. The green, light blue, yellow, and red arrows represent the positions of hinge domain, coiled-coil domain, head domain, and HEAT-repeat subunits, respectively. The purple arrow represents the position of the elbow on the coiled-coil domains. **(B)** The time trajectory of hinge angle. The blue and green lines represent data with and without median filter, respectively. The red arrows point to time frames where the angle exceeds 40°. **(C)** The frequency distribution of hinge angle. The data was fit by a linear combination (dark blue) of two Gaussian distributions (light blue).

previously observed by different methods [25,26,28]. Potentially due to the limited measurement accuracy, the two distributions show substantial overlap which prevent us from quantitating the relative stability of these two conformations. Note that the relative stability might vary depending on buffer conditions.

## AFM-fitting simulation to model the open hinge structure

To date, detailed structural information is available only for the closed hinge conformation (PDB: 4RSI) [18]. To obtain a structure for the open hinge conformation, we performed CG flexible-fitting MD simulations for an AFM image of an open hinge conformation (Fig 3A). For the fitting, we chose a single AFM image of the hinge with 79˚ hinge angle based on the assumption that DNA would tend to bind to the most open hinge. In each step of these simulations, the pseudo-AFM image was calculated from the simulation snapshot. Then, the forces derived from the difference between the pseudo- and the experimental AFM images was used to move the CG particles so that the pseudo-AMF image in the next step was more similar to the experimental image. Eventually, we obtained the structure which reproduces the experimental AFM image.

We obtained the initial structure by rigid-body rotation and translation of the closed hinge crystal structure (Fig 3B). In the subsequent flexible-fitting, the molecules, Smc2 and Smc4, are under the effect of the AFM fitting potential and the native-structure-based potential. These two potentials are to stabilize the open conformation observed in the AFM image and the closed conformation in the crystal structure, respectively. The $\kappa$ is the only controllable parameter that decides the stability balance between the open and closed conformation. The larger the value of κ is, the easier the fitting is, but the more distorted the subunit structures are. We varied κ as 10, 3000, and 10000. Each structure in the simulations were evaluated by three measures; inter-subunit Q-score, intra-subunit Q-score, and cosine similarity between the pseudo- and experimental AFM images. The Q-score is a standard nativeness score defined as the ratio of the number of bead-bead contacts formed at a certain moment in simulations to the number of contacts formed in the native protein structure. The inter-subunit Q-score measures similarity to the closed hinge structure, whereas the intra-subunit Q-score measures the structural integrity of each subunit.

Regardless of the $\kappa$ value, in most of the simulations, the Smc2 and Smc4 molecules diffused away from each other and got out of reach of the AFM potential (Fig 3C top; nearly zero inter-subunit Q-score). When κ was set to 10, the inter-subunit Q-score was larger than 0.8 in some simulations (6/240; Fig 3C top left). This result suggests that the hinge occasionally closes in the fitting simulations despite the presence of the AFM potential which stabilizes the open conformation. On the other hand, when κ was set to 10000, the inter-subunit Q-score ranged from 0.6 to 0.8 in some simulations (5/240; Fig 3C top right). The visual inspection confirmed that the structures with the inter-subunit Q-score in this range correspond to the open hinge. In this case, however, the intra-subunit Q-score is less than 0.95 in some simulations (6/240, Fig 3C bottom right), suggesting that the subunit structures are distorted. When κ was set to 3,000, the inter-subunit Q-score was in the range from 0.6 to 0.8 in a few simulations (2/240; Fig 3C top center). Also, the intra-subunit Q-score was almost always larger than 0.95 (239/ 240; Fig 3C bottom center). These results suggest that κ = 3000 is an appropriate value to stabilize the open hinge structure. We also tested κ = 1000 and 5000, and obtained similar results to those with κ = 3000. Thus, we set κ as 3000 for the rest of this study.

We plotted the time trajectories of the cosine similarity between the pseudo- and experimental AFM images (*c.s.* in Eq 2) and inter-subunit Q-score (Fig 3D and 3E). Fig 3D shows that *c.s.* increases right after the start of the simulation. Conversely, the inter-subunit Q-score takes longer until it suddenly reaches 0.7 (Fig 3E). These plots suggest that the two subunits are initially roughly positioned according to the AFM potential and subsequently, the inter-subunit contacts are recovered according to the native structure-based potential.

In the AFM image, Smc2 and Smc4 subunits could not be distinguished (Fig 3A). In the initial structure, Smc2 and Smc4 were placed in the region I and II, respectively, by a random

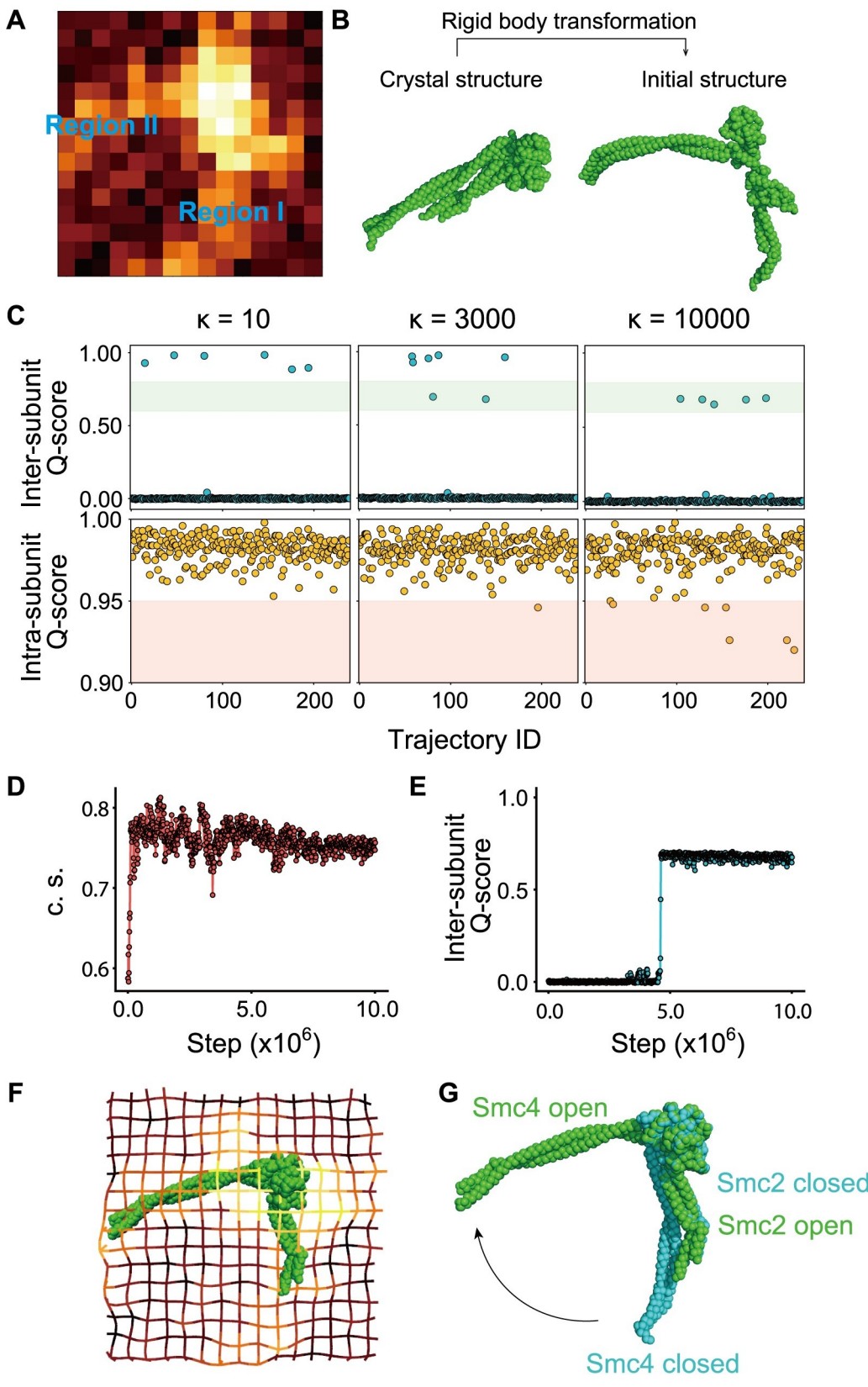

**Fig 3.** **(A)** The AFM image of the open hinge. One pixel corresponds to 1.875 nm. **(B)** The crystal hinge structure and the initial structure of AFM fitting simulations. The initial structure was obtained by a manual rigid body transformation of the crystal structure. **(C)** Inter-subunit Q-scores and intra-subunit Q-scores. Q-score is defined as the ratio of the number of bead-bead contacts formed at a certain moment in simulations to the number of contacts formed in the native protein structure. The scores were calculated from simulations with the fitting strength parameter κ set to 10, 3000, and 10000. **(D)** The time trajectory of the cosine similarity (c.s.) between an experimental AFM image and the pseudo-AFM image calculated from the simulations. The definition of c.s. is provided in the main text. **(E)** The time trajectory of intra-subunit Q-score. **(F)** The open hinge structure obtained with the AFM fitting simulation overlaid with a grid representation of the reference AFM image. **(G)** The crystal structure (light blue) superimposed on the simulated open hinge structure.

choice. When the position was exchanged (Smc2 in region II and Smc4 in region I), we obtained almost the same, but the upside-down structure. This result suggests that the AFM potential dictates the hinge angle but not the orientation of the molecule.

From our simulations, we obtained a structure for the open hinge conformation that reproduces the experimental AFM images (Fig 3F). Superposition of a part of globular domains (residue 505 to 676 of Smc2 and 671 to 835 of Smc4) of the open (simulated structure) and closed (crystal structure PDB ID: 4RSI) hinge structures illustrates that the Smc2 coiled-coil has the same orientation in both conformations (Fig 3G). Therefore, this result suggests that the Smc4 coiled-coil changes its orientation drastically upon the hinge conformational change.

## Prediction of dsDNA binding surfaces on the hinge domain

To predict dsDNA binding surfaces on the hinge domain, we performed CGMD simulations of dsDNA with the closed hinge structure (PDB:4RSI) and the open hinge structure obtained by the AFM fitting simulation (Fig 3F). During the simulation, we set κ to 3000 to stabilize the open hinge structure. We then placed five 18-bps dsDNA molecules randomly in the simulation box.

After starting the open and closed hinge simulations, we observed spontaneous dsDNA association with the hinge (Fig 4A and 4B). In the open hinge simulation, dsDNA bound both to the 'inside' region between the coiled-coils and to the opposite 'outside' region (Fig 4A and 4C). This result is consistent with previous suggestions for dsDNA and ssDNA binding surfaces based on the conservation of positively charged residues in the hinge domain of SMC proteins [18,24,25]. As expected, dsDNA could not access the inside region in the closed hinge simulation (Fig 4B and 4D). Instead, our simulation predicts that dsDNA weakly binds to two distinct sites on the outside region. The surface is accessible even in the folded-back state of the condensin holo-complex observed in the previous cryo-EM study [28]. The comparison of Fig 4C with Fig 4D suggests that these two binding sites in the closed hinge merge into one in the open hinge.

We calculated the electrostatic potential on the hinge surface using the APBS plugin of PyMol (https://pymol.org) to see the difference of electrostatic potential between the open and closed hinge structures (Fig 4E and 4F). Fig 4E (left) shows that positively charged surfaces inside the hinge are exposed upon the hinge opening. Comparison of Fig 4E (right) with Fig 4F (right) shows that two positively charged surfaces in the closed hinge merge into one region in the open conformation. Together, the hinge opening is required to make the inside dsDNA binding site accessible and to merge two small positively charged outside surfaces into a single large surface.

In this study, we performed CGMD simulations of Ycg1/dsDNA (Fig 1), Ycs4/dsDNA (S2 Fig), and hinge/dsDNA complexes (Fig 4). To compare the DNA binding strength predicted for the hinge domain to that predicted for the Ycg1 and Ycs4, we plotted the survival probabilities of DNA molecules bound to the different DNA-binding surfaces at 400 mM salt

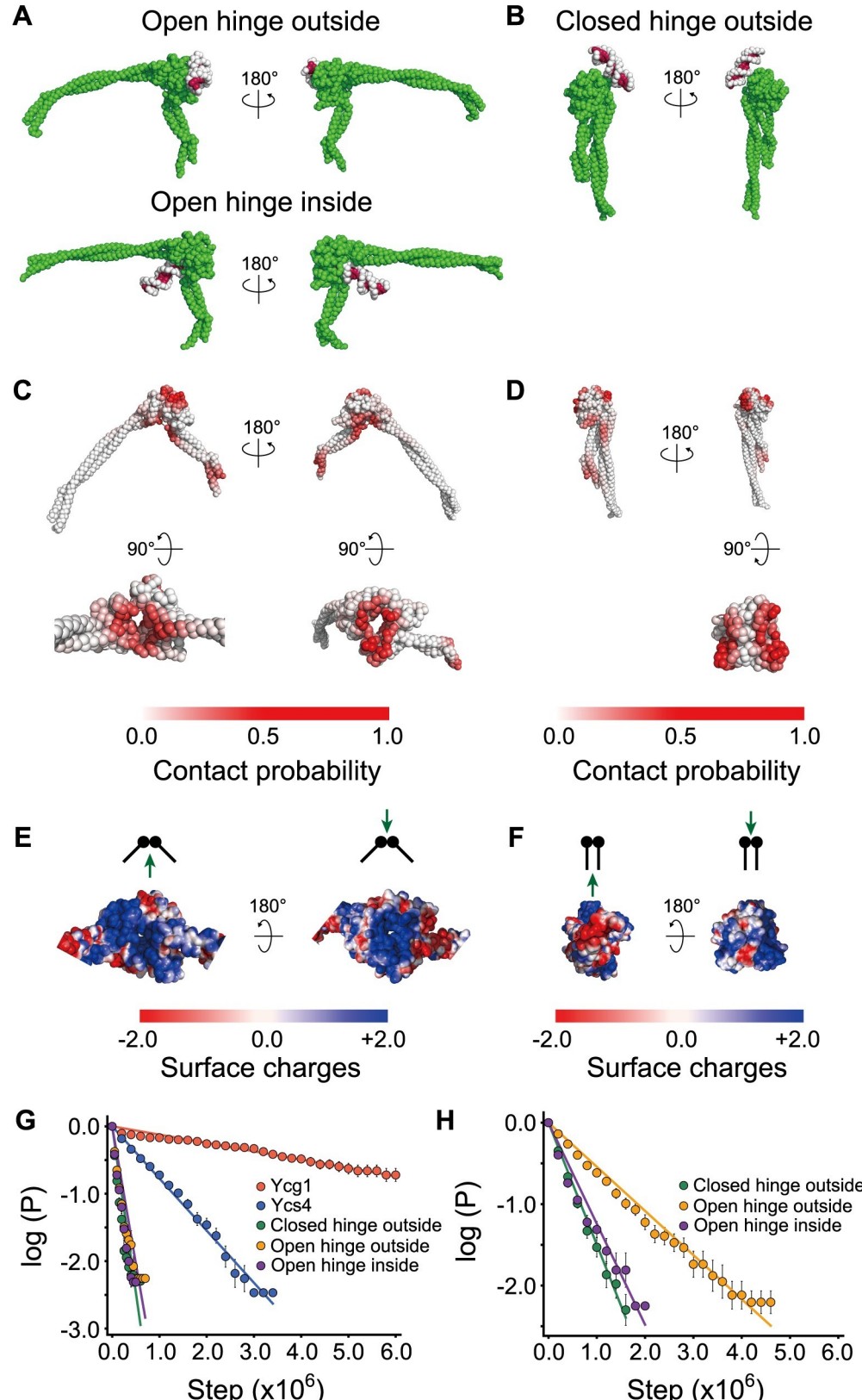

**Fig 4.** **(A, B)** The representative structures from the CGMD simulations of the open (A) and closed hinge (B) with dsDNA (the coordinates are available in S1 and S2 Data). **(C, D)** dsDNA contact probabilities mapped on the open (C)

and closed (D) hinge structures. Darker red particles contact with DNA with a higher probability. **(E, F)** Surface charges mapped on the open (E) and closed (F) hinge structures. **(G, H)** The survival probability plots for the Ycg1/dsDNA, Ycs4/dsDNA, and hinge/dsDNA binding in 400 (G) and 200 (H) mM salt concentration conditions, respectively. *P* represents the probability of DNA binding on the indicated region after a certain duration.

concentration (a weak binding condition) (Fig 4G). The simulations showed that dsDNA bound most strongly to Ycg1 (dissociation rate constant $1.2\times10^{-7}$ steps$^{-1}$), followed by Ycs4 ($7.7\times10^{-7}$ steps$^{-1}$) and the hinge domain (open outside $4.1\times10^{-6}$ steps$^{-1}$, closed outside $4.9\times10^{-6}$ steps$^{-1}$, open inside $5.4\times10^{-6}$ steps$^{-1}$). The fastest dissociation from the inside of the hinge domain suggests that, although dsDNA can potentially access the open hinge inside, any interaction might only be transient. The ability to detect transient DNA binding, which might be missed in biochemical and structural studies, is the strength of MD simulations.

At the 400 mM salt concentration, dsDNA binding at the hinge domain was too weak to compare the binding strengths between the different surfaces. Therefore, we repeated the simulation at 200 mM salt concentration for the open and the closed hinge with dsDNA to calculate the survival probabilities (Fig 4H). The result shows that dsDNA binding to the outside region of the open hinge ($5.4\times10^{-7}$ steps$^{-1}$) is slightly stronger than the inside region ($1.2\times10^{-6}$ steps$^{-1}$) or the outside region of the closed hinge ($1.4\times10^{-6}$ steps$^{-1}$), further confirming that dsDNA binding to the inside of the hinge domain might only be transient. Note that the dsDNA binding to any surface of the open and closed hinges is weaker than the binding to Ycg1.

We also tested whether the hinge can close while dsDNA remains bound to the inside. To accomplish this, we performed simulations at 10 mM salt concentration with the AFM fitting potential using the open hinge with dsDNA bound to the inside as an initial structure. Then, we performed simulations without the AFM fitting potential. In this simulation, the structure relaxes under the effect of the native-structure-based potential that stabilizes the reference closed hinge conformation. Thus, the hinge closed within $2.1\times10^{6}$ steps on average without dsDNA dissociation (79/80 simulations) (Fig 5A and 5B). Eventually, we obtained the closed hinge structure with dsDNA trapped inside.

Next, we performed simulations at 400 mM salt concentration using the closed hinge structure with dsDNA trapped inside as an initial structure and observed the closed hinge slid off from one end of dsDNA. The survival probability plots (Fig 5C) show that dsDNA binding to the closed hinge inside ($9.2\times10^{-8}$ step$^{-1}$) is significantly stronger than the open hinge inside ($5.4\times10^{-6}$ step$^{-1}$) and is comparable to the binding strength calculated for Ycg1 ($1.2\times10^{-7}$ step$^{-1}$). This result indicates that dsDNA can access the inside of the open hinge and might then induce the closing of the hinge, which considerably stabilizes dsDNA binding.

We then assessed the importance of amino acid residues on the inside of the hinge domain. We performed a PSI-BLAST search using the amino acid sequence of *Saccharomyces cerevisiae* condensin as a query and obtained a dataset containing 1000 related sequences from various species. We calculated a conservation score for each residue and plotted the distribution of these scores (Fig 6A). The frequency distribution of the conservation scores (Fig 6A) has two peaks, one around 0.0 and the other around 0.9. The amino acids with a high conservation score may play an essential role in the condensin function. The hinge structure in which each amino acid is colored according to its conservation score (Fig 6B) reveals that the positively charged amino acids (arginine and lysine) on the inside have relatively high conservation scores ($> 0.8$), fully consistent with the previous studies [22,24]. Thus, these residues likely have an essential role in dsDNA binding.

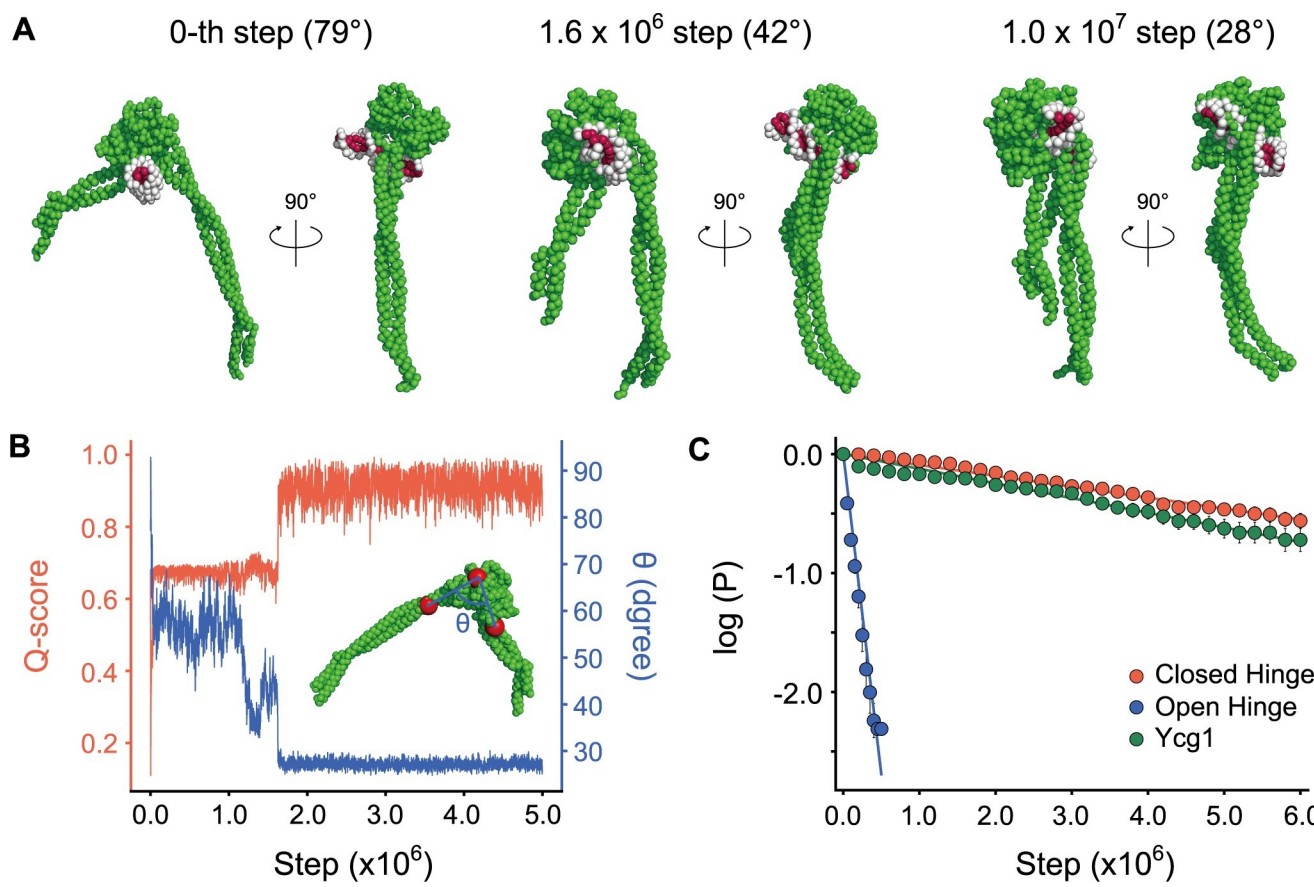

**Fig 5. (A)** The structures of the 0-th, $1.6 \times 10^6$-th, and $1.0 \times 10^7$-th steps of the CGMD simulation of the hinge/dsDNA complex without the AFM fitting potential. The structure relaxes under the effect of the native-structure-based potential that stabilizes the reference closed hinge conformation. **(B)** The time trajectories of inter-subunit Q-score and hinge angle from a CGMD simulation. The inset illustrates the hinge angle. The red and blue lines represent intra-molecule Q-score and hinge angle ($\theta$), respectively. **(C)** The survival probability plot for hinge/dsDNA and Ycg1/dsDNA binding. *P* represents the probability of DNA binding on the indicated region after a certain duration.

## Discussion

The purpose of this study is to predict the dsDNA binding surfaces on the condensin hinge domain, which has been suggested in literature [17,18,21–25]. In this study, we conducted HS-AFM imaging of the budding yeast condensin holo-complex and used this data as basis for CGMD simulations to model the hinge structure in a transient open conformation. We then simulated the dsDNA binding to open and closed hinge conformations, predicting that dsDNA binds to the outside surface when closed and to the outside and inside surfaces when open. The simulations also show that the coiled-coils might be able to close around dsDNA bound to the inside and thereby stabilize the dsDNA binding.

Ryu *et al.* have recently imaged condensin/dsDNA complex structures using AFM in a dry condition. The image (Fig 5A in [25]) indicated that dsDNA binds to the hinge domain. However, due to low resolution of the image, the structural detail of the hinge/dsDNA complex has been missing. Also, the crystal structure of the hinge domain [18] suggested that the inner surface of the hinge domain is inaccessible to dsDNA. Thus, the molecular mechanism by which dsDNA get access to the inner surface of the hinge domain has also remained unclear. The results of AFM imaging and simulations in this study proposed that the hinge domain repeats opening and closing and dsDNA can get access to the inner surface when the hinge opens. To

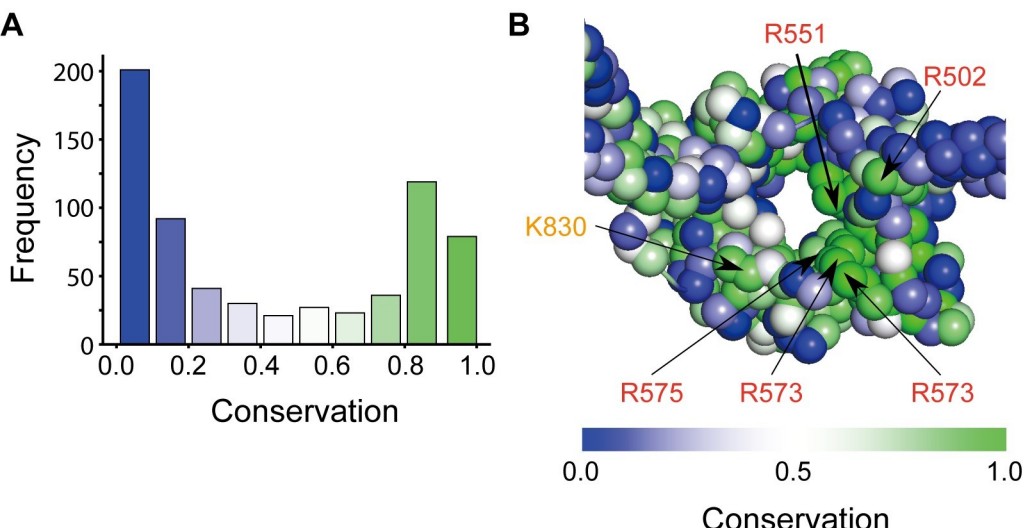

**Fig 6.** **(A)** The frequency distribution of conservation probability of each amino acid in the hinge domain (PDB ID: 4RSI). **(B)** The open hinge structure in which each particle is colored according to its conservation probability. The positively charged amino acids (arginine and lysine) with relatively high conservation scores ($> 0.8$) were labeled.

validate this model, future works should manage to observe dynamics of the condensin/DNA complex in solution using HS-AFM. Also, the current work found the positively charged amino acids likely having an essential role in dsDNA binding. The importance of these residues should also be tested in future.

Soh *et al.* showed that the yeast condensin hinge domain binds to dsDNA. However, they did not address the role of this binding *in vivo*. Recently, Nunez *et al.* created a mutant (3A-hng) in which three amino acids involved in the dsDNA binding of the bacterial SMC hinge were replaced with alanine [45]. The mutations do not significantly affect translocation of the bacterial SMC complex along dsDNA, indicating that the hinge is irrelevant for dsDNA translocation. However, it has not been explored if the dsDNA binding to the budding yeast condensin hinge is essential *in vivo*. Thus, *in vivo* experiments should be performed using mutant yeast condensin in the near future

Uchiyama *et al.* revealed that ssDNA binds to the outside surface of the hinge [24]. In this binding, the initially buried binding surface should be exposed by the hinge conformational change. Consistently, the current study suggested that dsDNA binds to the hinge outside. The *in vivo* function of the ssDNA and dsDNA binding should be investigated in the future.

In this study, we used CGMD simulations to predict the protein/dsDNA complex structures. The CGMD simulations increase the calculation speed with some compromise on accuracy compared with all-atom MD simulations. The CG model neglects the protein-dsDNA hydrophobic interaction, based on the assumption that the electrostatic interaction is dominant in the protein-dsDNA interaction [36]. However, the prediction could be more accurate if we introduce hydrophobic interactions into the CG model [46]. Also, the predicted CG structure could be refined using all-atom MD simulations [47]. It is expected that improvement of the CGMD simulation method will lead to a more accurate prediction of a protein/DNA complex structure.

Previous studies proposed the scrunching [25] and the loop capture models [31] (not mutually exclusive) as molecular mechanisms for the loop extrusion, both of which assume the dsDNA binding to the hinge domain. The simulations in this study support a model in which dynamic structural changes in the hinge regulate the dsDNA binding. The previous study

suggested that the structural change around the head domain induced by ATP hydrolysis is propagated along coiled-coils to the hinge domain and leads to hinge opening. Based on this model and results from the current study, we speculate that the ATP hydrolysis might control the binding of dsDNA to the hinge domain. Thus, the hinge closing and opening, and the accompanying dsDNA binding and dissociation might underlie the molecular mechanism of DNA-loop extrusion by condensin [31,48].

## Supporting information

**S1 Fig. Time trajectories of Q-score calculated from the Ycg1/DNA complex simulations.** The Q-score represents how much fraction of the protein/DNA contacts in the crystal structure forms in each simulation snapshot. We considered that protein and DNA particles contacted when these were within 15Å. DNA particle identity was neglected to make this measure robust for DNA sliding. The representative structures with Q-scores of 0.5, 0.8, and 1.0 (the crystal structure) were also presented.
(PDF)

**S2 Fig.** **(A)** The initial structure of CGMD simulations of the Ycs4/dsDNA complex. **(B)** The representative structures of the CGMD simulation. **(C)** DNA contact probabilities mapped on the Ycs4 structure.
(PDF)

**S3 Fig. The representative AFM images of the condensin holo-complex.** In these images, one of the HEAT repeat subunits pointed by the red arrow repeatedly dissociated from and associated to the head domains of Smc2 and Smc4.
(PDF)

**S1 Data. Coordinates for the hinge/DNA complex structure in the PDB format.**
(PDB)

**S2 Data. Coordinates for the hinge/DNA complex structure in the PDB format.**
(PDB)

## Acknowledgments

We thank members of the theoretical biophysics laboratory at Kyoto University for discussion and assistance throughout this work. We also thank Prof. Christian H. Haering for his comments and advice on the manuscript. We acknowledge the technical support for the HS-AFM from Prof. Toshio Ando and Prof. Takayuki Uchihashi.

## Author Contributions

**Conceptualization:** Hiroki Koide, Tsuyoshi Terakawa.

**Data curation:** Hiroki Koide, Noriyuki Kodera.

**Funding acquisition:** Shoji Takada, Tsuyoshi Terakawa.

**Investigation:** Hiroki Koide, Noriyuki Kodera, Tsuyoshi Terakawa.

**Methodology:** Tsuyoshi Terakawa.

**Resources:** Shveta Bisht.

**Software:** Shoji Takada.

**Supervision:** Shoji Takada, Tsuyoshi Terakawa.

**Writing – original draft:** Hiroki Koide, Tsuyoshi Terakawa.

**Writing – review & editing:** Noriyuki Kodera, Shveta Bisht, Shoji Takada, Tsuyoshi Terakawa.

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
