## [Decision Letter · Decision Letter 0]

4 Jun 2021

Dear Dr. Terakawa,

Thank you very much for submitting your manuscript "Modeling of DNA binding to the condensin hinge domain using molecular dynamics simulations guided by atomic force microscopy" for consideration at PLOS Computational Biology. As with all papers reviewed by the journal, your manuscript was reviewed by members of the editorial board and by several independent reviewers. The reviewers appreciated the attention to an important topic. Based on the reviews, we are likely to accept this manuscript for publication, providing that you modify the manuscript according to the review recommendations.

Sincerely,

Alexander MacKerell

Associate Editor

PLOS Computational Biology

Arne Elofsson

Deputy Editor

PLOS Computational Biology

[LINK]

Reviewer's Responses to Questions

**Comments to the Authors:**

Reviewer #1: This manuscript combines high-speed atomic force microscopy (hsAFM) and coarse grained molecular dynamics (cgMD) simulations to model double strand DNA (dsDNA) binding to the hinge domain of condensin, which is involved in DNA loop extrusion to compact chromatin during mitosis. The manuscript is well-written and well-organized, but I do have a couple of doubts/concerns detailed below.

The cgMD simulations sacrifice physical accuracy for length/time-scale expediency, so the initial sanity check of modeling the known closed hinge domain dsDNA-bound crystal structure is well-justified. However, the setup of those simulations and their analysis is not fully satisfying. The removal of Brn1 from the complex in the cgMD simulations changes the comparison, and even if it is necessary to enable dsDNA binding, it could have been included in separate cgMD simulations as a second randomly placed component to see whether the whole complex could be modeled correctly. More importantly, it is not clear to what extent the structure is modeled correctly in the cgMD simulations. The use of minimum distances and principal component analysis and DNA contacts obscures the simple question of how close the model is to the actual structure, which can be answered easily by comparing a coarse-grained RMSD between the model and the crystal structure. This RMSD should be reported as a time series for all 5 dsDNA molecules binding to the condension hinge domain for all cgMD simulations performed.

The other concern I have is the difficulty for a reader of this manuscript to judge what other structures are presently available for connectin hinge domains and how they relate to the cgMD simulations in this manuscript. I think the best way to address this is to show all the known structures in a side-by-side visual depiction in the coarse-grained format, and also to provide coarse-grained RMSD time series analysis for proximity to each known structure for the cgMD simulations.

Finally, the coarse-grained 3D coordinates for a predicted model for the DNA-bound open hinge domain should be provided in the supplementary material.

Reviewer #2: The manuscript presents a computational study on some structural aspects of the SMC and predicts the binding between condensin and DNA. A coarse-grained model is used which is calibrated against AFM data. The study contributes to this interesting topic and is very timely. It is clearly written and the figures are elegant. There are some methodological aspects that are not clear that require some clarifications to make the manuscript easy to read and to apply its approach to other systems.

1. In the modeling of the interactions between Ycg1 and DNA (Figure 1), the protein seems to be of the same conformations. Is it modeled as a flexible molecule? How much the flexibility affects its binding to DNA?

2. In the modeling of the open hinge, how many AFM images were included? Was the fitting of the computational model done to a single AFM image?

3. In the modeling of the open hinge structure, the modeling using kappa is unclear and more details may help. Was this transition modeled just by a single parameter? Also, choosing the value of kappa is unclear. The differences between the results for k=10, 3000, 10000 are very small (Figure 3C). The difference between them are of just less than 10 simulations while the other >200 simulations behave very similarly? Are the simulations sensitive to kappa? Also in figure 3C, k should be changed to kappa.

4. In general, the adding more details to the figure legends may help to understand the figures.

5. In some simulations, the interactions between the DNA was studied with open or closed hinge. However, Figure 5 presents a transition between the closed to open. How was this modeled? Also with kappa? If so, how sensitive this to the value of kappa?

**Have the authors made all data and (if applicable) computational code underlying the findings in their manuscript fully available?**

Reviewer #1: **No: **The coarse grained 3D coordinates of the DNA-bound open hinge domain model should be available as a pdb file in the supplementary material.

Reviewer #2: Yes

PLOS authors have the option to publish the peer review history of their article (what does this mean?). If published, this will include your full peer review and any attached files.

Reviewer #1: No

Reviewer #2: No

Figure Files:

Data Requirements:

Reproducibility:

References:

---

## [Editor Report · Decision Letter 1]

10 Jul 2021

Dear Dr. Terakawa,

We are pleased to inform you that your manuscript 'Modeling of DNA binding to the condensin hinge domain using molecular dynamics simulations guided by atomic force microscopy' has been provisionally accepted for publication in PLOS Computational Biology.

Best regards,

Alexander MacKerell

Associate Editor

PLOS Computational Biology

Arne Elofsson

Deputy Editor

PLOS Computational Biology

---

## [Editor Report · Acceptance letter]

22 Jul 2021

PCOMPBIOL-D-21-00831R1 

Modeling of DNA binding to the condensin hinge domain using molecular dynamics simulations guided by atomic force microscopy

Dear Dr Terakawa,

I am pleased to inform you that your manuscript has been formally accepted for publication in PLOS Computational Biology. Your manuscript is now with our production department and you will be notified of the publication date in due course.

With kind regards,

Olena Szabo
